# Bioinformatic Characterization of the Functional and Structural Effect of Single Nucleotide Mutations in Patients with High-Grade Glioma

**DOI:** 10.3390/biomedicines12102287

**Published:** 2024-10-09

**Authors:** Sara Vélez Gómez, Juliana María Martínez Garro, León Darío Ortiz Gómez, Jorge Emilio Salazar Flórez, Fernando P. Monroy, Ronald Guillermo Peláez Sánchez

**Affiliations:** 1Faculty of Sciences and Biotechnology, CES University, Medellín 050021, Colombia; savelez@ces.edu.co; 2CES Biology, Science and Biotechnology School, CES University, Medellín 050021, Colombia; jmartinezg@ces.edu.co; 3Cancer Institute, Las Americas-AUNA Clinic, Medellín 050023, Colombia; leonortizgomez@gmail.com; 4GEINCRO Research Group, Medicine Program, School of Health Sciences, San Martín University Foundation, Sabaneta 055457, Colombia; jorge.salazarf@sanmartin.edu.co; 5Department of Biological Sciences, Northerm Arizona University, Flagstaff, AZ 85721, USA; fernando.monroy@nau.edu; 6Life and Health Sciences Research Group, Graduate School, CES University, Medellín 050021, Colombia

**Keywords:** neoplasia of the central nervous system, computational analysis, protein, variant, *TP53*, *IDH1*, *IDH2*

## Abstract

**Background:** Gliomas are neoplasms of the central nervous system that originate in glial cells. The genetic characteristics of this type of neoplasm are the loss of function of tumor suppressor genes such as *TP53* and somatic mutations in genes such as *IDH1/2*. Additionally, in clinical cases, de novo single nucleotide polymorphisms (SNP) are reported, of which their pathogenicity and their effects on the function and stability of the protein are known. **Methodology:** Non-synonymous SNPs were analyzed for their structural and functional effect on proteins using a set of bioinformatics tools such as SIFT, PolyPhen-2, PhD-SNP, I-Mutant 3.0, MUpro, and mutation3D. A structural comparison between normal and mutated residues for disease-associated coding SNPs was performed using TM-aling and the SWISS MODEL. **Results:** A total of 13 SNPs were obtained for the *TP53* gene, 1 SNP for *IDH1*, and 1 for *IDH2*, which would be functionally detrimental and associated with disease. Additionally, these changes compromise the structure and function of the protein; the A161S SNP for *TP53* that has not been reported in any databases was classified as detrimental. **Conclusions:** All non-synonymous SNPs reported for *TP53* were in the region of the deoxyribonucleic acid (DNA) binding domain and had a great impact on the function and stability of the protein. In addition, the two polymorphisms detected in *IDH1* and *IDH2* genes compromise the structure and activity of the protein. Both genes are related to the development of high-grade gliomas. All the data obtained in this study must be validated through experimental approaches.

## 1. Introduction

High-grade gliomas are primary tumors of the central nervous system (CNS), which, according to the fifth classification of the World Health Organization (WHO) 2021, are classified according to their histological and molecular characteristics into astrocytomas, oligodendrogliomas, and glioblastomas [1]. Patients diagnosed with this neoplasia have a poor prognosis, with only 20% of them surviving around 5 years. More specifically, the two patients diagnosed with glioblastomas were expected to survive about 2 years (14.8% patients) and 10 years (2.6%), Amut-IDH, sin-codel-1p/19q, G3-4, and these patients were expected to survive about 2 years (43.3%) and 10 years (19% patients), Omut-IDH, codel-1p/19q, G3, and these patients are expected to survive about 2 years (68.6% patients) and 10 years (39.3% patients). Despite aggressive treatment with surgery, chemotherapy, and radiotherapy [2,3], oncological diseases arise because of genetic alterations in genes that control critical biological processes. The accumulation of mutations over time allows the survival and progressive transformation of abnormal cell populations that eventually lead to tumor formation. In the case of gliomas, this process is known as gliomagenesis, which is characterized by several biological events, such as activated growth factor, damage to receptor signaling pathways, negative regulation of many apoptotic mechanisms, and imbalance between pro-angiogenic and antiangiogenic factors [4].

Establishing the process of gliomagenesis is of vital importance to identify the underlying molecular mechanisms involved in tumor progression. Several central signaling pathways have been identified in this process: one of these is the RTK/RAS/PI3K, TP53, and RB1 pathway, where TP53 mutations are critical for gli-oma progression [5]. TP53 plays a key role in maintaining cellular homeostasis and is frequently deregulated in oncological pathologies such as gliomas. The protein is positioned at the center of a regulatory network that controls cell proliferation, survival, genome integrity, and other functions. As a transcriptional regulator, TP53 integrates stress signals and promotes cell cycle arrest, senescence, and apoptosis to prevent damaged cells from spreading [6]. The TP53 mutational status is associated with glioblastoma progression [7] and TP53 inactivation is correlated with a more invasive [8], less apoptotic [9], more proliferative, and stem cell-like phenotype [10]. Glioblastoma cell lines that possess inactive mutated TP53 are more resistant to DNA-damaging therapeutic drugs, such as cisplatin.

Within the process of gliomagenesis, IDH1/2 mutations have been identified as frequently occurring in the early stages of human glioma development. IDH mutation has been shown to promote glioma development through DNA and histone methylation [11]. In a study of 1010 diffuse gliomas, IDH1 mutations were detected in 70.9% of tumors, while IDH2 mutations were observed in 3.1% of tumors [11]. These two mutations give rise to neomorphic enzymatic activity, leading to distinct patterns in cancer metabolism, epigenetic alterations, and resistance to therapy. The mutant enzymes, in particular, the IDH1 R132H and IDH2 R172K variants, have been widely studied in low- and high-grade gliomas. These variants are characterized by conferring a specific enzymatic activity that converts alpha-ketoglutarate to 2-hydroxyglutarate; the accumulation of this oncometabolite in gliomas with IDH mutations profoundly affects several cellular processes [12], including metabolic reprogramming, because the Krebs cycle adjusts to compensate for fluctuations in the pathways. An analysis of metabolic flux suggested that cells with IDH mutations exhibit increased oxidative metabolism in the Krebs cycle, while the reductive metabolism of glutamine is suppressed [13]. The objective of this study was to evaluate the functional and structural effects generated by mutations in the genes that encode the TP53 and IDH1/2 proteins in a cohort of 31 patients with high-grade glioma.

## 2. Materials and Methods

### 2.1. Study Population

The sample analyzed was obtained from patients who attended the Cancer Institute of the Clínica Las Américas-AUNA Medellín-Colombia between 2019 and 2021 to receive treatment for high-grade gliomas. Of the total population, 31 patients participated by open invitation with a diagnosis of glioblastoma, oligodendroglioma, or astrocytoma.

### 2.2. Samples for Sequencing

We collected a total of 26 blocks of paraffin-embedded brain tissue and 5 liquid biopsies. From these samples, DNA was extracted and used for the sequencing of 324 genes involved in tumor processes, thus obtaining the mutations for the 31 patients with high-grade gliomas.

### 2.3. Molecular Mutation Panel

Genomic sequencing was performed through the Foundation one^®^ CDx (F1CDx) panel, which is a diagnostic methodology based on next-generation sequencing for the detection of mutations such as substitutions, insertions, copy number alterations (CNA) in 324 selected genes, and gene rearrangements, as well as genomic fingerprints including microsatellite instability, tumor mutational burden, and loss of heterozygosity. They used the results of the sequencing process to perform univariate, bivariate, and multivariate analyses to identify mutated genes that were related to the gliomagenesis process. We identified significant differences in the group of 324 sequenced genes. The genes (PIK3C2B, ERBB3, KIT, and MLH1) were found to have significant differences. However, the genes *TP53* and *IDH1/2* were chosen for this research due to their importance in the gliomagenesis process, biological plausibility, and the criteria of the investigators, despite not having significant differences between the group of 324 sequenced genes.

### 2.4. Prediction of Pathogenic SNPs

To predict the changes in the structure and function of the protein produced by the SNPs, we used the SIFT (Sorting Intolerant From Tolerant) software (https://sift.bii.a-star.edu.sg/, accessed on 21 June 2024) [14] and PolyPhen-2 (prediction of functional effects of human nsSNPs (http://genetics.bwh.harvard.edu/pph2/, accessed on 21 June 2024) [15]. PolyPhen-2 stratifies the SNPs into likely damaging, probably damaging, or benign, and generates a score between 0 and 1 where ≤0.85 means a greater probability of damage. As for SIFT, it generates two categories (tolerated/neutral) where values ≤ 0.05 are classified as damaging.

### 2.5. Prediction of Deleterious SNPs for the Protein

We used the servers for the PhD-SNP (Predictor of human Deleterious Single Nucleotide Polymorphisms) (https://snps.biofold.org/phd-snp/phd-snp.html, accessed on 21 June 2024) and SNPs and GO (predicting disease-associated variations using GO terms) (https://snps.biofold.org/snps-and-go/snps-and-go.html, accessed on 21 June 2024). Both predictors determine whether SNPs can cause disease. For example, PhD-SNP classifies nsSNPs of a gene into neutral or human disease-causing mutations [16], while SNPs and GO predict disease-associated variations using GO terms [17]; the harmful nonsynonymous SNPs predicted were analyzed with these two tools to determine their association with high-grade gliomas, and these were classified as diseases or neutral according to their potential to cause disease. SNPs labeled as a disease were retained for further analysis. The reliability index (RI) in PhD-SNP, SNPs, and GO ranges from 0 to 10, where 10 means the highest reliability.

### 2.6. Predicting Protein Stability for Functionally Deleterious Glioma-Associated SNPs

The SNPs involved in the gliomagenesis process were further analyzed to determine their effects on the protein stability of the bioinformatics programs I-Mutant 3.0 (https://folding.biofold.org/i-mutant/i-mutant2.0.html, accessed on 21 June 2024) and MUpro (https://mupro.proteomics.ics.uci.edu/, accessed on 21 June 2024); the first one is an algorithm based on an SVM (Support Vector Machine), which automatically predicts the change in the stability of the proteins after a single nucleotide mutation, while the I-Mutant 3.0 algorithm was trained with a data set derived from Protherm [18] that provides an estimate of the change in the Gibbs free energy (Delta Delta G or DDG), calculated by subtracting the DDG value of the non-mutated protein from the DDG value of the mutated protein (unit kcal/moles). Having a DDG > 0 means higher protein stability, while a DDG < 0 refers to lower protein stability. The MUpro server not only calculates using an SVM but also uses a neural network. It gives a prediction value between −1 and 1, where a value < 0 indicates a decrease in stability [19].

### 2.7. Prediction of Protein Structural Alteration and Loss of Activity

To predict structural changes in different proteins, including alteration in activity and binding, the MutPred2 web application (http://mutpred.mutdb.org/, accessed on 21 June 2024) was used, which is a method and software package based on machine learning that integrates genetic and molecular data to probabilistically reason about the pathogenicity of amino acid substitutions. It is trained on a set of 53,180 pathogenic and 206,946 unlabeled variants obtained from the Human Gene Mutation Database (HGMD), SwissVar, dbSNP, and pairwise alignment between species that can predict more than 50 protein features [20]. The application provides different probability percentages for each of the characteristics; this probability varies between 0 and 1, where the score closest to 1 indicates that it is more likely that the mutation alters the property of the protein, generating a gain or loss of the function [20]. We applied this tool to explore aspects of the activity of the TP53 and IDH proteins and how these are affected by SNPs.

### 2.8. Observation of Amino Acid Change

The Project-HOPE database (https://www3.cmbi.umcn.nl/hope/, accessed on 21 June 2024) provides information on amino acid changes in protein structures along with changes in different physiological and chemical activities that occur after mutation [21]. From this interface, it is possible to report whether the residue is in a conserved site, interpret mutations that are most likely to be harmful, and therefore significantly affect the function of the protein. We analyzed the SNPs of the *TP53* and *IDH1/2* genes in this database to obtain information on the produced changes.

### 2.9. Structural Comparison between Normal and Mutated Residues

The TM-Align application (https://zhanggroup.org/TM-align/, accessed on 21 June 2024) is an algorithm for sequence-independent protein structure comparisons. It gives a score between 0 and 1, where 1 indicates a perfect match between the mutated and wild-type protein structures, while a score ≥ 0.5 indicates that the same fold/topology exists between the structures located in the SCOP/CATH databases. which divides the protein into domains and classifies them at the hierarchical level. For example, SCOP classifies domains into classes, folds, superfamilies, and families. For its part, the four levels in CATH are class, architecture, topology, and homologous superfamily, thus generating a comparison between the structures [22]. Scores of 0.2 correspond to unrelated proteins chosen at random [23]. This tool was used to perform an analysis to establish a structural difference and thus conclude that SNPs affect the protein. Root means square deviation or RMSD scores were also provided, which measures the average distance between atoms of overlapping molecules, where higher scores indicate more structural differences between the two compared molecules and scores closer to 0 indicate greater geometric similarity between the two residues. The SWISS-MODEL program (https://swissmodel.expasy.org/, accessed on 21 June 2024) models the three-dimensional structure of proteins and provides the Ramachandran plot score between two structures [24]. The two values, both the TM and the Ramachandran graph score, helped determine the structural differences caused by the alterations.

### 2.10. Group of Mutations

SNPs for TP53 that were classified as deleterious by the tools described above were analyzed using the Mutation3D software (http://mutation3d.org/, accessed on 21 June 2024), which predicts and visualizes the spatial arrangement of mutated amino acids in the protein structure, showing whether they are found in an important domain. It can also describe clusters of mutations to discover some more significant SNPs [25].

### 2.11. Protein–Protein Interaction

The interaction of TP53 and IDH1/2 proteins with other proteins was studied using the STRING web server (https://string-db.org, accessed on 21 June 2024), which predicts the main proteins that show interactions with the query gene. STRING predicts such interactions based on gene fusion, co-expression, function, and experimental data. It shows combined scores for each interacting protein, ranging from 0 to 1, where 0 shows the lowest interaction and 1 indicates the highest interaction [26]; in addition, the KEEG database (https://www.genome.jp/kegg/, accessed on 21 June 2024) was used to identify the metabolic pathways associated with the *TP53* and *IDH1*/2 genes.

## 3. Results

### 3.1. Distribution of SNPs

From 31 patients, we selected the genes with the highest mutation frequency from the Foundation one ^®^ panel; therefore, 13 SNPs were cataloged as non-synonymous for the *TP53* gene and 1 for the *IDH1* gene. Polymorphism for IDH2 was selected for its biological importance in the development of gliomas (Table 1).

### 3.2. Unique Non-Synonymous Polymorphisms Predicted Using SIFT and PolyPhen-2

The PolyPhen-2 and SIFT software (https://sift.bii.a-star.edu.sg/, accessed on 21 June 2024) were run and thresholds were determined using the MSC (mutation significance cutoff) program available at (https://www.hgid.org/computational-tools/, accessed on 21 June 2024). The MSC for TP53 was (PolyPhen2 = 0.033 and SIFT = 0.133), IDH1 (PolyPhen2 = 0.823 and SIFT = 0.995), and IDH2 (PolyPhen2 = 1.000 and SIFT = 0.995). This score was used for later analysis. We determined, with the PolyPhen-2 tool, that 13 non-synonymous SNPs for the *TP53* gene were damaging, while for the SIFT software, only 11 were classified as harmful. The non-synonymous SNPs (A161S, C124T) showed scores greater than 0.00 and are reported as tolerable mutations (Table 1). The same happened with the IDH1 SNP, which is tolerated in SIFT and damaging for PolyPhen-2. For the IDH2 polymorphism, this was harmful for both predictors.

### 3.3. Disease-Related Single Nucleotide Polymorphisms PhD-SNP SNPs and GO

The two programs identified the 13 SNPs for TP53 as damaging for the protein, as well as the SNPs that had conflicting interpretations (C124T and A161S) (Table 1). Therefore, they were considered harmful and were further analyzed bioinformatically. In the case of IDH1/2, both SNPs were predicted to cause protein alteration; therefore, no SNP was excluded from the analysis.

### 3.4. Prediction of Protein Stability Alteration Using I-Mutant 3.0 and MUpro Software

Of the 13 single nucleotide polymorphisms for TP53, 12 had a DDG value < 0, showing less protein stability because of the mutation. In the case of the non-synonymous SNPs of the *IDH1/2* genes, in both cases, the software predicted that the mutations decreased the protein stability. Reliability index scores are provided along with the DDG value to demonstrate the decrease in structural stability of the proteins. Changes classified as harmful not only cause damage to the phenotype but also reduce the overall stability of the protein (Table 2).

### 3.5. Mutations Group

SNPs that were predicted to be damaging, disease-associated, and causing decreased protein stability were further analyzed with mutation3D software (https://www.mutation3d.org/, accessed on 21 June 2024) to predict clusters of mutations. A total of 13 SNPs were analyzed for TP53. The analysis showed that the clustered SNP positions are marked in red and occupy the DNA binding domain of the protein (Figure 1a); conversely, positions 124, 125, and 279 are not included in these frequent mutation sites. Therefore, they are marked in blue. Additionally, it is shown how the groups of mutations are in the DNA binding domain. In the case of IDH1, the R132H mutation is not found in the sites of high mutation frequency, but it is in one of the protein domains (Figure 1b). For the IDH2 protein, the R172K polymorphism was in sites of low mutational frequency, which is why it is marked in blue. Additionally, it is also located in one of the protein domains.

### 3.6. Impact of Single Nucleotide Polymorphisms on Protein Structural Activity

For the 13 non-synonymous SNP of TP53 and the 2 of IDH1/2 that were suggested as deleterious, we carried out a functional characterization of each of the amino acids, in both the wild type and mutated form using the Project HOPE database. For the Y236C mutation of the *TP53* gene, the wild-type amino acid differs in size with respect to the mutant, which will cause an empty space in the body of the protein. The mutated residue is more hydrophobic; therefore, the change will produce the loss of hydrogen bonds in the body of the protein, resulting in altered folding. As for the V274G mutation, it will cause a loss of hydrophobic interactions in the body of the protein; in addition, this mutation will generate an empty space because the mutated residue is smaller (Figure 2). The main difference in the V157F change is the size between the mutated residue and the wild type since there is a probability that the mutated amino acid does not fit at a spatial level. For the C238W change, the normal residue is 100% conserved for this position, so a change in this position would result in damage to the protein. For the A161S mutation, the residues differ in size; additionally, there is a loss of hydrophobic interactions in the body of the protein. The T125M mutation causes the loss of hydrophobicity; therefore, there are fewer hydrogen bonds in the body of the protein, which could alter protein folding. The R248Q and R248W changes are characterized by having a difference in charge between the wild type and the mutated amino acid, so interactions with other molecules could be lost. The wild-type amino acid is not conserved in the case of the Y220H mutation; however, the mutation will cause an empty space in the body of the protein, and a difference in hydrophobicity is also present. The R175H alteration is identified by generating changes in charge between residues [27]. Both C124T and H179Q changes differ in size between amino acids; therefore, an empty space will be generated that may result in the loss of external interactions with other molecules. Regarding the G279E change, the wild-type amino acid has a neutral charge, unlike the mutated, which has a negative charge, which can cause problems in the proper folding of the protein.

For its part, the R132H mutation found in the IDH1 protein predicts that the mutated residue is smaller, and its charge is neutral, unlike the wild type, which has a positive charge, which will prevent its interaction with other molecules that, in this specific case, could not interact with isocitrate (Figure 2). For the R172K mutation located in the IDH2 protein, the wild-type amino acid is 100% conserved, so changes in it could be harmful to the protein. Additionally, the mutated residue is smaller, which produces an empty space in the body of the protein.

### 3.7. Alteration and Loss of Activity

We further analyzed SNPs using the MutPred2 web server to calculate alterations in protein activity (Table 3). According to the analysis, we found that Y220H, R175H, and H179Q polymorphisms of the TP53 protein mainly affected the metal binding property, while changes in the Y236C, V274G, and V157F altered the stability of the protein (Table 3). For mutations A161S, T125M, R248Q, and R248W no results were obtained. On the other hand, for the IDH1 protein, we found that this R132H mutation, which presented changes in metal binding, also impacted the allosteric site and loss of relative accessibility to the solvent (Table 3). Regarding the R172K mutation for the IDH2 protein, it presents changes in activity such as loss of the catalytic site and metal and DNA binding sites (Table 3).

### 3.8. Structural Differences between Normal and Mutated Residues Using TM-Aling and Ramachandran Plots

We ran the TM-aling and SWISS-MODEL programs to further analyze the changes between the structures of the wild-type proteins and the homology-modeled mutated proteins. The lower the TM score and the higher the RMSD value, the greater the difference between the structures. On the contrary, it may also be the case that the TM scores may be high as well as the RMSD value. Furthermore, the higher the score of the Ramachandran graph, the more the structure will be favored (Table 4). It was shown that the C238W polymorphism for TP53 affects the structure of the protein, while for the IDH1 protein, it was found that R132H was the most deleterious SNP, having a low TM score and higher RMSD, with a significant score in favor of the Ramachandran plot. This scenario is different from the non-synonymous SNP of IDH2; it has a high TM score and a very low RMSD score.

### 3.9. Protein–Protein Interaction

The search in the STRING database resulted in the direct interaction of TP53 with 10 proteins, with confidence scores (>0.95). These results were obtained according to our experiments, data sets, and text mining (Figure 3). All these proteins predicted strong functional associations with TP53 and were shown to be involved in the activation of different pathways such as cell cycle regulation, DNA damage response, metabolism, apoptosis, and autophagy. The analysis of the interaction network shows that CREBBP, MDM2, and EP300 proteins contribute to the co-activation of TP53 as a transcription factor. The protein–protein interaction analyzed for the IDH1 and IDH2 isoenzymes showed the presence of 10 proteins with confidence scores between 0.99 and 0.97, all of them were involved in the oxidative decarboxylation process of isocitrate (Figure 3). They are involved in the metabolic pathway of tricarboxylic acids generated in the mitochondria.

## 4. Discussion

In the development of gliomas, a significant number of biological processes are compromised in which genes such as *TP53* and *IDH1/2* are involved. The comprehensive characterization of the non-synonymous SNPs that may be present in the coding regions allows us to elucidate the relationship between the genetic variants and their functional and structural impact on a protein, which can lead to discovering possible markers for the diagnosis and prognosis of this type of neoplasms.

The bioinformatics analysis found that the 13 polymorphisms reported for the TP53 gene were detrimental since they modify the structure and function of the protein. Each of the variants participates in the p53-ARF-MDM2 metabolic pathway, which is mutated in 84% of glioblastomas. This pathway is related to processes of cell invasion, migration, and proliferation, as well as to evasion of apoptosis [6]. All SNPs were in the DNA binding site, which is considered a site of high mutation frequency, given that molecular profiling of various tumors has shown that most of the 190 amino acids found in this protein domain are mutated in one or more cancers, and, in many cases, are homozygous in the tumor. Therefore, they are believed to contribute to the development of neoplasia [28]. These mutations are found at different frequencies and in various oncological diseases; for example, in the case of the SNP rs730882026 (Y236C), different databases such as ClinVar (https://www.ncbi.nlm.nih.gov/clinvar/, accessed on 21 June 2024) and Varsome (https://varsome.com/, accessed on 21 June 2024) report this variant with conflicting interpretation, unlike the results obtained where it is shown that it is detrimental. In the literature, it has been associated with lung cancer, has not been associated with high-grade gliomas [29,30], and has been related to resistance to the chemotherapeutic agent Crizotinib^®^ [31]. For its part, the variant rs1057520006 (V274G) generates a loss of protein function; despite this, it has a conflicting interpretation in the databases. This mutation was classified as detrimental in this study.

Somatic mutations in TP53 are typically single-nucleotide missense changes that allow the production of a full-length protein. Some of these mutated proteins not only lose tumor suppressor function but can also acquire oncogenic properties, including increased invasion, proliferation, and chemo-resistance. This is the case for the rs121912654 (V157F) variant associated with lung cancer and hepatocellular carcinoma, where the mutation predicted a worse outcome for the patient [32,33]. According to ClinVar, its interpretation is not clear; however, the variant in this study was detrimental. For the SNP rs193920789 (C238W), its interpretation appears to be detrimental, as do our obtained results, where it is predicted that it is one of the variants that modify the structure and function of the protein the most. There are no reports in the literature indicating a detrimental association with this polymorphism or its clinical significance.

In our study, we report on a polymorphism (A161S) not yet classified in the databases. This represents an SNP with conflicting interpretation since there are predictors, such as SIFT that classify it as tolerable. Therefore, experimental validation is required, e.g., an in-vitro functional assay, to determine whether this SNP impacts protein function. For its part, the rs786201057 (T125M) variant has shown that the mutated protein has deficiencies in transactivation activity [34] and has also been related to diseases such as Li-Fraumeni syndrome [35]. This variant can be classified as detrimental because predictors and other programs associate it with a loss of protein structure and function. The polymorphisms (R248Q and R248W) are considered sites of high mutation frequency; several studies have shown that these changes promote cell migration and proliferation, through AKT signaling [36]. The R248 is mutated in three amino acids, R248Q, R248W, and R248L, and it has been shown that the first change induces more aggressive tumors in mice compared with other mutation hotspots [37]. The second change acquires neomorphic tumorigenic functions, particularly in invasion and the metastasis of solid tumors [38]. The R248Q mutation has mainly been associated with ovarian cancer and myeloid leukemia, while R248W is associated with pancreatic and lung cancer [39]. The SNP rs530941076, classified as detrimental, is related to different types of cancer such as breast, lung, and liver [40]. This change produces a slight alteration of intramolecular interactions which causes a loss of DNA binding [41].

One of the *TP53* gene variants reported with high frequency in oncological diseases is R175H. It is at a critical point for the structure, characterized by lower thermodynamic stability [42]. This SNP is classified as detrimental according to our results. This mutation has been associated with colorectal, breast, lung, pancreas, and myeloid leukemia [43]. In the specific case of gliomas, this has been related to a worse prognosis for the patient and rapid cancer progression. In high-grade glioma, recurrent TP53 point mutations may be the key to tumor progression, thus emphasizing its importance in gliomagenesis [44]. The C124T polymorphism is located deep in the DNA binding domain; however, it does not play a significant role in modulating the DNA binding affinity of TP53 [45]. In this study, this SNP provided conflicting interpretations because the predictors were unable to conclude whether it was detrimental, and the rest of the web servers did not conclude it modified the structure and function of the protein.

In the different databases, the G279E variant is reported with conflicting interpretations. In this study, this SNP was detrimental; however, the stability of the protein was increased, which could be related to the gliomagenesis process. Finally, the H179Q SNP is classified as detrimental; cells that express this mutated protein are immortal and show attenuation of the G1 checkpoint function, loss of expression of the p21 protein [46], and it is related to osteosarcomas and melanomas [47].

The *TP53* gene has a key role in several cellular processes. It often undergoes mutations, which facilitate the appearance of oncological disease. Its role in gliomas is related to the onset of neoplasia, for example, in 30% to 40% of astrocytoma cases, a loss of TP53 function can be observed at an early stage [48]. Additionally, mutations in this gene are related to low-grade gliomas, unlike this study where all patients were adults and were diagnosed with high-grade gliomas, which confirms that these mutations may be an early-onset event.

The R132H mutation in IDH1 is considered an early event in gliomagenesis. At a clinical level, glioma patients carrying mutant IDH1 respond better to antitumor therapies. However, the mechanism by which mutations in this gene contribute to gliomagenesis and therapeutic response has remained difficult to elucidate [49]. The presence of R132H in glial cells significantly enhanced cell senescence in response to temozolomide and radiation through a DNA damage-mediated mechanism. The R132H change occurs heterozygously in the tumor since the homozygous form produces an inactive protein. The biological impact of the mutated protein in oncogenesis is related to the state of homeostasis in the Krebs cycle. Glutamine and/or Glutamate serve as key substrates to offset the metabolic impact by promoting lipid and glutathione synthetic pathways. Gliomas with IDH1 mutation show a different metabolic pattern compared to other solid tumors [13]. Regarding treatment, many studies have shown that gliomas with IDH mutations respond better to standard therapeutic methods such as temozolomide and radiotherapy [50]. Bioinformatic characterization studies of the R132H SNP have revealed that it is generated at a critical point in the protein [51]. This mutation resulted in conflicting interpretations for the pathogenicity predictors, but at the same time, this mutation was the one that most modified the structure of the protein (see Table 4). It is found in more than 95% of secondary glioblastomas and in less than 10% of primary glioblastomas, being classified as a definitive marker for this type of neoplasm [52].

The R172K mutation in the IDH2 enzyme is related to different malignancies, including gliomas, chondrosarcomas, and myeloid leukemia, among others. This mutation confers neoactivity to the enzyme and confers epigenetic alterations to the DNA [53]. The implications of this mutation in the structure of the enzyme are related to its conformational state since its natural state is to be in an open conformation, but when it binds to isocitrate, it adopts a closed conformation and properly assembles the active site for the normal reaction to take place [54]. Therefore, R172K is expected to affect enzyme activity; the IDH2 variants are less common, with R172K observed in 3% of glioma patients, and detection of the R172K variant has implications for glioma diagnosis, prognosis, and potential treatment [55]. In this study, the variant was pathogenic, which agrees with the databases and affects the function and structure of the protein.

Regarding mutations that lead to the loss of binding of P53 and IDH1/2 proteins to metal ions, the following have been described: the P53 protein depends on certain endogenous metals for its function, especially on Zinc (Zn) ions [56]. Structural studies have demonstrated the presence of Zn ions in its active site [57] and the importance of these metal ions in the following functions has been verified: binding of the DNA-binding domain (DBD) to DNA occurs only in the presence of Zn; the absence of Zn decreases the activity of the protein; the DNA binding domain is stabilized by Zn; the interaction between P53–DNA is altered by mutations in the DNA binding domain; mutations in the DBD domain lead to misfolding of the protein and poor binding to Zn ions; mutations in the DBD domain lead to the poor transcriptional activity of the P53 protein; and in the absence of Zn, binding to DNA is considered less stable. Additionally, it has been discovered that Zn can help reactivate mutated proteins and that Zn sequestration leads to the inactivation of the P53 protein [58,59,60,61,62,63,64,65,66,67,68,69,70]. This highlights the importance of P53 mutations (Y220H = 27%, R175H = 67%, and H179Q = 72%), which alter, to a certain extent, the binding of metals to the P53 protein and could lead to all these alterations in the function of the protein.

On the other hand, it has been observed that divalent metal ions such as magnesium and manganese (Mg2 and Mn2), when binding to the substrate (isocitrate) of the IDH1 protein, enhance the catalytic activity of the protein [71]. The R132H mutation represents about 80% of the mutations found in this gene and reduces the activity of the wild-type IDH1. But it confers a gain of function in the mutated proteins to produce oncometabolites such as (2-hydroxyglutarate) [x16]. The IDH1 mutation (R132H = 55%), by reducing the metal binding of the protein, could affect the catalytic activity of the wild-type IDH1 by converting isocitrate to α-ketoglutarate in the cytoplasm.

The association between *IDH2* gene mutations and metal ions has been little explored. However, due to the similarity in amino acid sequences, shared domains, and three-dimensional structure with respect to IDH1, it could be suggested that their metabolic activity would be affected in a similar way. The IDH2 mutation (R172K = 40%), by reducing the metal binding of the protein, could affect the catalytic activity of the wild-type IDH2 by converting isocitrate to α-ketoglutarate in the mitochondria.

All the single nucleotide polymorphisms for the TP53 and IDH1/2 proteins discussed in this manuscript are associated with promoting the development of glioma tumorigenesis since it has been shown that the proteins have some effect both in their level of activity and in their structure. The combination of these mutations has not been associated with patient survival results [72], so more correlation studies are needed between the described mutations and their outcome. The combination of new bioinformatic methods with next-generation sequencing is part of daily clinical care and an essential tool for the diagnosis of low- and high-grade gliomas, providing more precise information, which can help reduce the need for additional surgeries, and providing important information for additional radiotherapy or chemotherapy treatments depending on the diagnosis. Despite these advances, it is important to note that there is still much to be done in this area, particularly in the development of innovative and improved bioinformatics models for so-called translational research.

## 5. Conclusions

Computational analysis and identification of deleterious polymorphisms of TP53 and IDH1/2 in patients with high-grade gliomas are essential to distinguish differences in the properties of mutated and wild-type proteins. The collection of this data is applicable to the diagnosis and prognosis of this type of patient and may additionally indicate treatments based on the genetics of the tumor. More clinical research is needed where the association of these polymorphisms with the severity of the neoplasia can be studied.

## Figures and Tables

**Figure 1 biomedicines-12-02287-f001:**
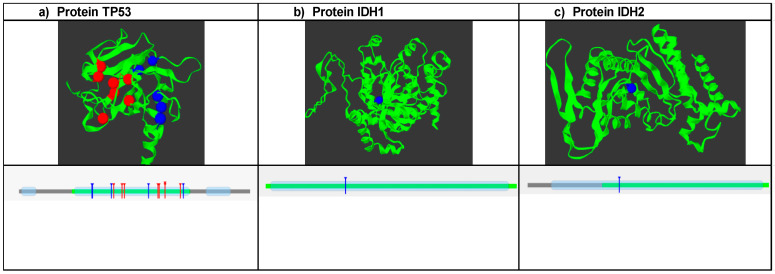
Analysis of mutation clusters using Mutation 3D software (https://www.mutation3d.org/, accessed on 21 June 2024). (**a**) Shows the structure of the protein and the location of frequent mutation sites marked in red, and the blue dots are mutations that are outside of these; the lower graph shows the position of the mutation clusters in the domains of the TP53 protein. (**b**) Structure of the IDH1 protein: where the R132H mutation is represented, it is located outside of the frequent mutation sites, and the lower graph shows the position of the polymorphism in the protein domain. (**c**) IDH2 protein and the position of the R172K mutation are shown, which are located within a domain of this.

**Figure 2 biomedicines-12-02287-f002:**
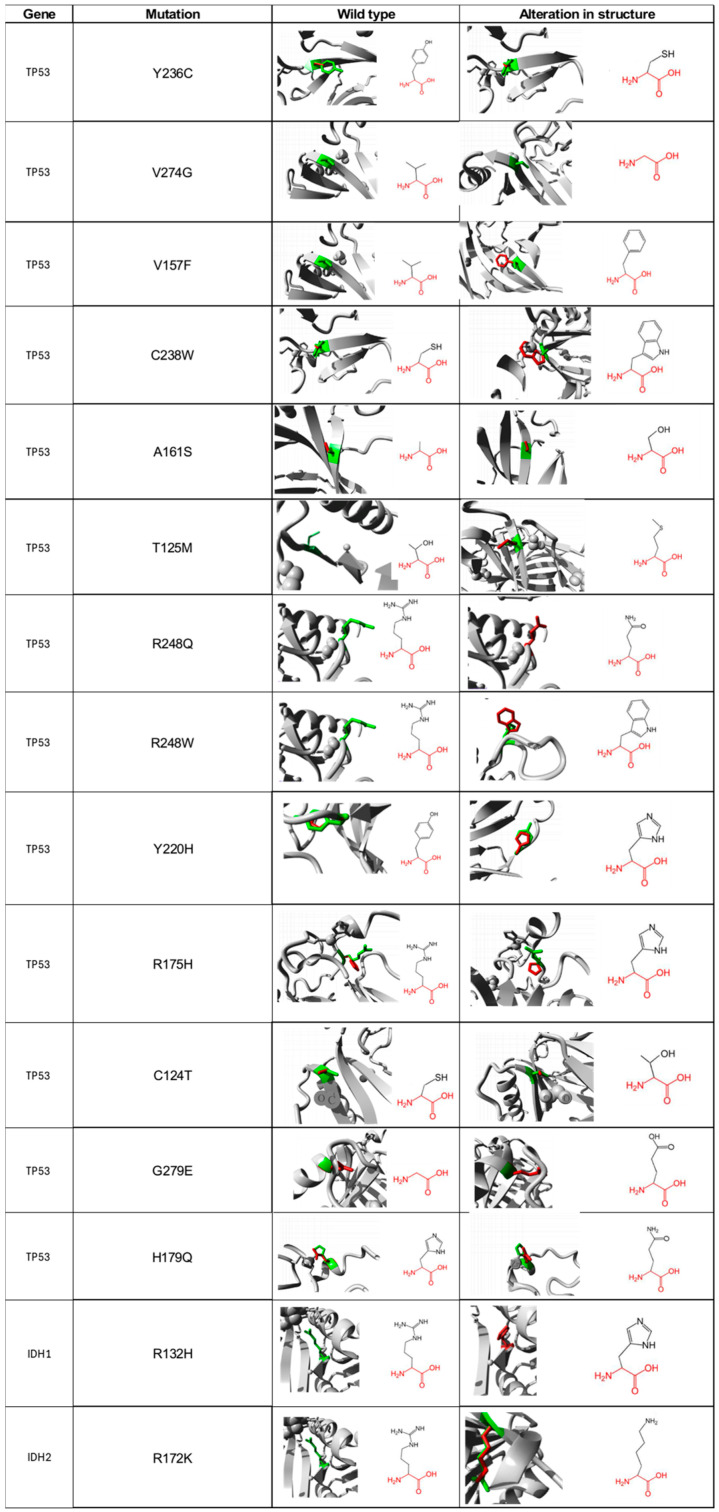
Visual representation of amino acid structural changes caused by SNPs in the *TP53* and *IDH1/2* genes. Data taken from the Project-HOPE web server, a total of 13 deleterious SNPs were identified for *TP53*, 1 for *IDH1*, and 1 for *IDH2*.

**Figure 3 biomedicines-12-02287-f003:**
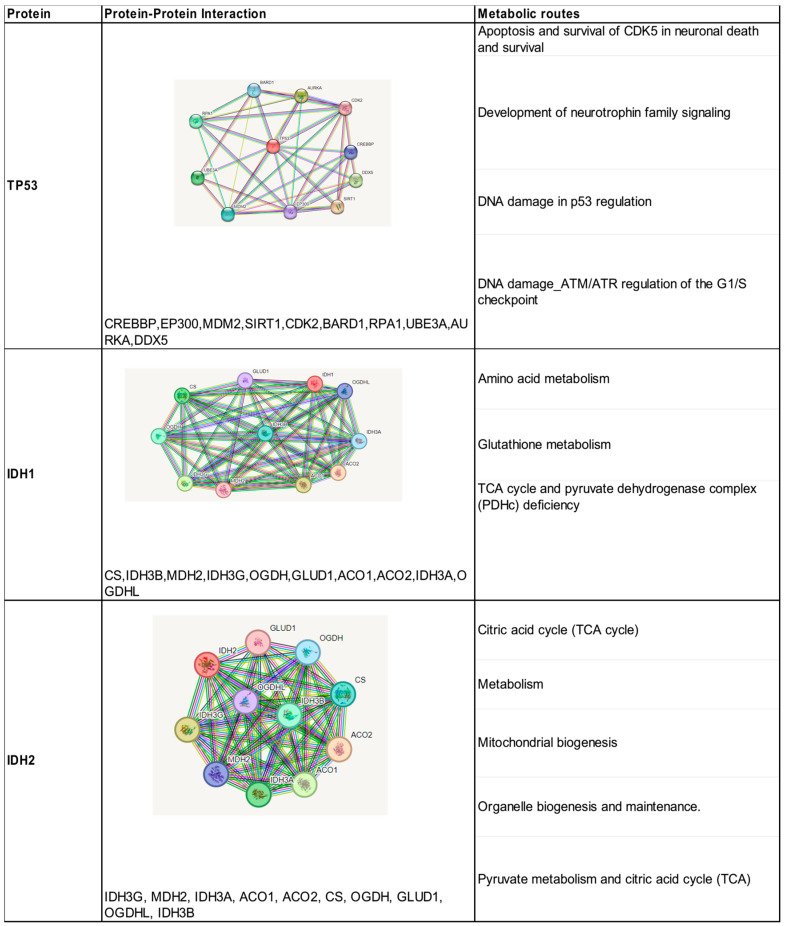
Identification of protein–protein interactions and metabolic pathways of the proteins encoded by the three mutated genes associated with high-grade gliomas.

**Table 1 biomedicines-12-02287-t001:** List of 15 non-synonymous single nucleotide polymorphisms for the *TP53* and *IDH1/2* genes. * Mutation not reported in databases such as ClinVar.

Chromosome	Mutation Position	RS	Gene	Amino Acid Alteration	Classification According to ClinVar	MSC/ PolyPhen Score	Prediction PolyPhen	MSC/SIFT Score	Prediction	PhD SNP	SNP and GO
									SIFT		
17	7577574	rs730882026	*TP53*	p.Tyr236Cys	Conflict of interpretation	0.033/1	Pathogenic	0.133/0	Pathogenic	Deleterious to protein	Deleterious to protein
17	7577117	rs1057520006	*TP53*	p.Val274Gly	Conflict of interpretation	0.033/1	Pathogenic	0.133/0	Pathogenic	Deleterious to protein	Deleterious to protein
17	7578461	rs121912654	*TP53*	p.Val157Phe	Conflict of interpretation	0.033/0.999	Pathogenic	0.133/0.01	Pathogenic	Deleterious to protein	Deleterious to protein
17	7577107	rs193920789	*TP53*	p.Cys238Trp	Pathogenic	0.033/1	Pathogenic	0.133/0	Pathogenic	Deleterious to protein	Deleterious to protein
17	7578449	*	*TP53*	p.Ala161Ser	New	0.033/0.967	Pathogenic	0.133/0.08	Tolerable	Deleterious to protein	Deleterious to protein
17	7577086	rs786201057	*TP53*	p.Thr125Met	Conflict of interpretation	0.033/1	Pathogenic	0.133/0	Pathogenic	Deleterious to protein	Deleterious to protein
17	7577538	rs11540652	*TP53*	p.Arg248Gln	Pathogenic	0.033/1	Pathogenic	0.133/0	Pathogenic	Deleterious to protein	Deleterious to protein
17	7577539	rs121912651	*TP53*	p.Arg248Trp	Pathogenic	0.033/1	Pathogenic	0.133/0	Pathogenic	Deleterious to protein	Deleterious to protein
17	7578191	rs530941076	*TP53*	p.Tyr220His	Pathogenic	0.033/1	Pathogenic	0.133/0	Pathogenic	Deleterious to protein	Deleterious to protein
17	7578406	rs28934578	*TP53*	p.Arg175His	Pathogenic	0.033/0.881	Pathogenic	0.133/0	Pathogenic	Deleterious to protein	Deleterious to protein
17	7579315	rs1555526478	*TP53*	p.Cys124Ter	Pathogenic	0.033/0.694	Pathogenic	0.133/0.11	Tolerable	Deleterious to protein	Deleterious to protein
17	7577102	rs1064793881	*TP53*	p.Gly279Glu	Conflict of interpretation	0.033/1	Pathogenic	0.133/0	Pathogenic	Deleterious to protein	Deleterious to protein
17	7578393	rs876660821	*TP53*	p.His179Gln	Pathogenic	0.033/0.999	Pathogenic	0.133/0	Pathogenic	Deleterious to protein	Deleterious to protein
2	209113112	rs121913500	*IDH1*	p.Arg132His	Pathogenic	0.823/0.013	benign	0.995/0	Pathogenic	Deleterious to protein	Deleterious to protein
**15**	90631838	rs121913503	*IDH2*	p.Arg172Lys	Pathogenic	1.000/0.999	Pathogenic	0.995/0	Pathogenic	Deleterious to protein	Deleterious to protein

**Table 2 biomedicines-12-02287-t002:** Analysis performed with I-Mutant 3.0 and MUpro that indicate the DDG and predict whether it increases or decreases the stability of the protein; the reliability index ranges from 1 to 10 and ensures that the value or prediction made is within the estimated interval, in this case, the G279E polymorphism has a conflict of interpretation between the two programs.

Gene	Alteration	MUpro		I-Mutant 3.0	
		Value DDG	Protein Stability	Reliability Index	Protein Stability
		(Kcal/mol)			
*TP53*	Y236C	−15,011.602	Decreases	3	Decreases
*TP53*	V274G	−25,962.627	Decreases	9	Decreases
*TP53*	V157F	−0.94054347	Decreases	9	Decreases
*TP53*	C238W	−14,075.569	Decreases	5	Decreases
*TP53*	A161S	−0.73476729	Decreases	9	Decreases
*TP53*	T125M	−0.22237917	Decreases	3	Decreases
*TP53*	R248Q	−18,402.017	Decreases	7	Decreases
*TP53*	R248W	−14,246.049	Decreases	5	Decreases
*TP53*	Y220H	−0.85387287	Decreases	7	Decreases
*TP53*	R175H	−29,105.592	Decreases	8	Decreases
*TP53*	C124T	−29,105.592	Decreases	1	Decreases
*TP53*	G279E	0.062230681	Increases	3	Decreases
*TP53*	H179Q	−0.19938582	Decreases	7	Decreases
*IDH1*	R132H	−14,231.866	Decreases	7	Decreases
*IDH2*	R172K	−12,425.191	Decreases	5	Decreases

**Table 3 biomedicines-12-02287-t003:** Protein activity data, structural alteration, and loss of binding from MutPred2, altered properties for TP53 SNPs, altered properties for non-synonymous SNP of the IDH1 protein, and altered properties for the IDH2 protein polymorphism, where probabilities close to 1 have a greater possibility that the mutation is affecting the protein property.

Altered Properties Due to SNP	SNPs Gene *TP53*								
	Percentage Probability of Change in Protein Function								
	Y236C	V274G	V157F	C238W	Y220H	R175H	C124T	G279E	H179Q
**Altered ordered interface**	28%	-	-	27%	-	-	28%	-	-
**Loss of strand**	27%	-	-	-	28%	-	-	27%	-
**Gain of disulfide bond at C238**	22%	-	-	26%	-	-	-	-	-
**Altered stability**	16%	73%	11%	-	-	51%	13%	-	-
**Loss of N-linked glycosylation at N239**	3%	-	-	6%	-	-	-	-	-
**Gain of intrinsic disorder**	-	42%	-	-	-	-	-	-	-
**Gain of strand**	-	-	-	26%	-	-	-	-	-
**Altered metal binding**	-	-	-	-	27%	67%	-	-	72%
**Gain of factor B**	-	-	-	-	24%	-	-	-	-
**Altered transmembrane protein**	-	-	-	-	14%	-	-	-	-
**Loss of sulfation at Y220**	-	-	-	-	1%	-	-	-	-
**Loss of helix**	-	-	-	-	-	28%	-	-	-
**Gain of ubiquitylation at K120**	-	-	-	-	-	-	15%	-	-
**Gain of helix**	-	-	-	-	-	-	-	30%	-
**Gain of intrinsic disorder**	-	-	-	-	-	-	-	31%	-
**Altered Properties Due to SNP**			**SNPs Gene *IDH1***						
			**Percentage Probability of Change in Protein Function**						
			**R132H**						
**Altered metal binding**			55%						
**Loss of allosteric site at R132**			36%						
**Altered DNA binding**			34%						
**Altered disordered interface**			29%						
**Altered transmembrane protein**			27%						
**Loss of relative solvent accessibility**			27%						
**Altered ordered interface**			26%						
**Gain of catalytic site at H133**			26%						
**Altered Properties Due to SNP**			**SNPs Gene *IDH2***						
			**Percentage Probability of Change in Protein Function**						
			**R172K**						
**Altered DNA binding**			40%						
**Loss of allosteric site at H173**			34%						
**Altered metal binding**			32%						
**Gain of relative solvent accessibility**			30%						
**Altered ordered interface**			29%						
**Loss of catalytic site at R172**			26%						
**Altered transmembrane protein**			22%						

**Table 4 biomedicines-12-02287-t004:** TM-matching score, RMSD value, and Ramachandran plot of identified SNP in *TP53*, *IDH1*, and *IDH2*. ^a^ a score of 1 indicates a perfect match between mutated and wild-type protein structures; ^b^ a higher RMSD score means more structural differences between the two compared proteins; ^c^ the Ramachandran plot score is calculated in percentages and provides an idea of how well two structures seem to compare to each other.

Gene	nsSNP	TM-Aling		SWISS-MODEL
		TM-Score ^a^	Value RMSD ^b^	Ramachandran Graphic Score ^c^
*TP53*	Y236C	1.00	0.003	95.40%
*TP53*	V274G	1.00	0.004	95.40%
*TP53*	V157F	1.00	0.041	95.40%
*TP53*	C238W	0.999	0.053	93.86%
*TP53*	A161S	1.00	0.001	95.40%
*TP53*	T125M	1.00	0.002	95.40%
*TP53*	R248Q	1.00	0.002	95.40%
*TP53*	R248W	1.00	0.001	95.40%
*TP53*	Y220H	1.00	0.002	95.40%
*TP53*	R175H	1.00	0.008	95.40%
*TP53*	C124T	1.00	0.006	95.40%
*TP53*	H179Q	1.00	0.002	95.40%
*TP53*	G279E	1.00	0.004	95.40%
*IDH1*	R132H	0.8382	3.730	96.33%
*IDH2*	R172K	1.00	0.004	96.44%

## Data Availability

Data are available in the NCBI database under the following accession numbers: TP53-Y236C (2874534), TP53-V274G (2874538), TP53-V157F (2874541), TP53-C238W (2874549), TP53-A161S (2874551), TP53-T125M (2874552), TP53-R248Q (2874554), TP53-R248W (2874556), TP53-Y220H (2874558), TP53-R175H (2874572), TP53-C124T (2874573), TP53-G279E (2874575), TP53-H179Q (2874578), IDH1-R132H (2874579), and IDH2-R172K (2874580).

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
