# Peer review of "Bioinformatic Characterization of the Functional and Structural Effect of Single Nucleotide Mutations in Patients with High-Grade Glioma"

_biomedicines, 2024, doi:10.3390/biomedicines12102287_

Round 1

Reviewer 1 Report

Comments and Suggestions for Authors

My comments and suggestions are as follows.

·        Page 1, Line 24-25: The study claims that "the pathogenicity and structural effects of SNPs are unknown," but several databases (ClinVar, COSMIC) already contain information on SNP pathogenicity. Consider revising to acknowledge existing data and emphasize the novelty of your bioinformatics approach.

·        Page 3, Line 45-47: The statement about a "poor prognosis" is a general observation about gliomas. Provide more detailed statistics (e.g., 5-year survival rates) to reinforce the point and offer readers clearer context.

·        Page 4, Line 56-58: The authors mention mutations in TP53, IDH1, and IDH2 but do not provide references to the most recent studies about their roles in gliomagenesis. Adding more recent and relevant citations would strengthen the literature review.

·        Page 5, Line 89-90: The panel of 324 genes used in sequencing is comprehensive but lacks detail regarding the rationale for the specific gene selection. Consider including a brief explanation of how these genes were selected and their relevance to glioma.

·        Page 7, Line 96-101: The bioinformatics tools used for SNP prediction (SIFT, PolyPhen-2) are widely accepted but prone to false positives/negatives. Including a discussion of their limitations and the validation steps taken (if any) would enhance the scientific rigor of this section.

·        Page 7, Line 109-110: The use of I-Mutant 3.0 and MUpro for stability predictions is helpful, but the manuscript would benefit from discussing any experimental validation of these computational findings, as in silico results can differ from in vitro data.

·        Page 8, Line 140-145: In the structural comparison between normal and mutated residues, the manuscript should provide more detailed data on the magnitude of changes in protein function (e.g., quantitative metrics for TM-score/RMSD) to give clearer insights into the biological impact of these mutations.

·        Page 10, Line 198-203: The manuscript concludes that certain SNPs in TP53 and IDH1/2 are "deleterious," but the threshold for determining this isn’t explained. Specify how cutoff values for PolyPhen-2, SIFT, and other tools were chosen for defining deleteriousness.

·        Page 12, Line 285-287: The manuscript mentions alterations in metal-binding properties due to SNPs but lacks an explanation of how these alterations mechanistically affect protein function. Provide more insight into how these changes contribute to gliomagenesis or tumor progression.

·        Page 14, Line 355-357: The mutation A161S is described as novel, but its pathogenic potential is conflicting. It would be beneficial to suggest potential experimental validation steps, such as in vitro functional assays, to determine whether this SNP indeed impacts protein function. 

Author Response

  • Page 1, Line 24-25: The study claims that "the pathogenicity and structural effects of SNPs are unknown," but several databases (ClinVar, COSMIC) already contain information on SNP pathogenicity. Consider revising to acknowledge existing data and emphasize the novelty of your bioinformatics approach.

The reviewer is right, all mutations are known, in fact, table 1 - column 6 shows the effect of the mutations found in the patients in the ClinVar database.

Therefore, we correct the sentence as follows:

Additionally, in clinical cases, de novo single nucleotide polymorphisms (SNP) are reported, of which their pathogenicity, and how they affect the function and stability of the protein are known.

  • Page 3, Line 45-47: The statement about a "poor prognosis" is a general observation about gliomas. Provide more detailed statistics (e.g., 5-year survival rates) to reinforce the point and offer readers clearer context.

The reviewer is right, more detailed statistics are provided:

Patients diagnosed with this neoplasia have a poor prognosis, with only 20% of them surviving around 5 years. more specifically, the two patients diagnosed with glioblastomas were expected to survive about two years (14.8% patients) and ten years (2.6%), Amut-IDH, sin-codel-1p/19q, G3-4, and these patients were expected to survive about two years (43.3%) and ten years (19% patients), Omut-IDH, codel-1p/19q, G3, and these patients are expected to survive about two years (68.6% patients) and ten years (39.3% patients). Despite aggressive treatment with surgery, chemotherapy, and radiotherapy (2, 20).

  • Page 4, Line 56-58: The authors mention mutations in TP53, IDH1, and IDH2 but do not provide references to the most recent studies about their roles in gliomagenesis. Adding more recent and relevant citations would strengthen the literature review.

The reviewer is right, so we add the following paragraph to the introduction:

Establishing the process of gliomagenesis is of vital importance to investigate the underlying molecular mechanisms involved in tumor progression. Several central signaling pathways have been identified in this process, one of these being RTK/RAS/PI3K, TP53 and RB1, where TP53 mutations are critical for glioma progression [5]. TP53 plays a central role in maintaining cellular homeostasis and is frequently deregulated in oncological pathologies such as gliomas. The protein is positioned at the center of a regulatory network that controls cell proliferation, survival, genome integrity and other functions. Working as a transcriptional regulator, TP53 integrates stress signals and promotes cell cycle arrest, senescence and apoptosis to prevent damaged cells from spreading [6]. The TP53 mutational status is associated with glioblastoma progression [7] and TP53 inactivation is correlated with a more invasive [8], less apoptotic [9], more proliferative and stem cell-like phenotype [10]. Glioblastoma cell lines that possess inactive mutated TP53 are more resistant to DNA-damaging therapeutic drugs, such as cisplatin.

Within the process of gliomagenesis, IDH1/2 mutations have been identified as frequently occurring in the early stages of human glioma development. Although IDH mutation has been shown to promote glioma development through DNA and histone methylation [11]. In a study of 1010 diffuse gliomas, IDH1 mutations were detected in 70.9% of tumors, while IDH2 mutations were observed in 3.1% of tumors [11]. These two mutations give rise to neomorphic enzymatic activity, leading to distinct patterns in cancer metabolism, epigenetic alterations and resistance to therapy. The mutant enzymes, in particular the IDH1 R132H and IDH2 R172K variants, have been widely studied in low- and high-grade gliomas. These variants are characterized by conferring a specific enzymatic activity that converts alpha-ketoglutarate to 2-hydroxyglutarate, the accumulation of this oncometabolite in gliomas with IDH mutations profoundly affects several cellular processes [12], including metabolic reprogramming, because the Krebs cycle adjusts to compensate for fluctuations in the pathways. An analysis of metabolic flux suggested that cells with IDH mutations exhibit increased oxidative metabolism in the Krebs cycle, while reductive metabolism of glutamine is suppressed [13]. The objective of this study was to evaluate the functional and structural effects generated by mutations in the genes that encode the TP53 and IDH1/2 proteins in a cohort of 31 patients with high-grade glioma.

References

  1. Luo J, Junaid M, Hamid N, Duan JJ, Yang X, Pei DS. Current understanding of gliomagenesis: from model to mechanism. Int J Med Sci. 2022;19(14):2071-9.

  1. Zhang Y, Dube C, Gibert M, Cruickshanks N, Wang B, Coughlan M, et al. The p53 Pathway in Glioblastoma. Cancers. 1 de septiembre de 2018;10(9):297.

  1. Krex D, Mohr B, Appelt H, Schackert HK, Schackert G. Genetic Analysis of a Multifocal Glioblastoma Multiforme: A Suitable Tool to Gain New Aspects in Glioma Development. Neurosurgery. 1 de diciembre de 2003;53(6):1377-84.

  1. Djuzenova CS, Fiedler V, Memmel S, Katzer A, Hartmann S, Krohne G, et al. Actin cytoskeleton organization, cell surface modification and invasion rate of 5 glioblastoma cell lines differing in PTEN and p53 status. Exp Cell Res. enero de 2015;330(2):346-57.

  1. Park CM, Park MJ, Kwak HJ, Moon SI, Yoo DH, Lee HC, et al. Induction of p53-mediated apoptosis and recovery of chemosensitivity through p53 transduction in human glioblastoma cells by cisplatin. Int J Oncol [Internet]. 1 de enero de 2006 [citado 17 de septiembre de 2024]; Disponible en: http://www.spandidos-publications.com/10.3892/ijo.28.1.119

  1. Zheng H, Ying H, Yan H, Kimmelman AC, Hiller DJ, Chen AJ, et al. p53 and Pten control neural and glioma stem/progenitor cell renewal and differentiation. Nature. octubre de 2008;455(7216):1129-33.

  1. Chen X, Liu J, Li Y, Zeng Y, Wang F, Cheng Z, et al. IDH1 mutation impairs antiviral response and potentiates oncolytic virotherapy in glioma. Nat Commun [Internet]. 25 de octubre de 2023 [citado 17 de septiembre de 2024];14(1).

  1. Grimi A, Bono BC, Lazzarin SM, Marcheselli S, Pessina F, Riva M. Gliomagenesis, Epileptogenesis, and Remodeling of Neural Circuits: Relevance for Novel Treatment Strategies in Low- and High-Grade Gliomas. Int J Mol Sci. 16 de agosto de 2024;25(16):8953.

  1. Han S, Liu Y, Cai SJ, Qian M, Ding J, Larion M, et al. IDH mutation in glioma: molecular mechanisms and potential therapeutic targets. Br J Cancer. 26 de mayo de 2020;122(11):1580-9.

  • Page 5, Line 89-90: The panel of 324 genes used in sequencing is comprehensive but lacks detail regarding the rationale for the specific gene selection. Consider including a brief explanation of how these genes were selected and their relevance to glioma.

The evaluator is right, therefore we include the following paragraph in the methodology section.

The results of the sequencing process were used to perform univariate, bivariate and multivariate analyses to identify mutated genes that were related to the glioma genesis process and significant differences between the group of 324 sequenced genes and in which mutations were identified. The genes (PIK3C2B, ERBB3, KIT, and MLH1) were found to have significant differences. However, the genes TP53 and IDH1/2 were chosen for this research due to their importance in the glioma genesis process, biological plausibility and the criteria of the researchers, despite not having significant differences between the group of 324 sequenced genes.

  • Page 7, Line 96-101: The bioinformatics tools used for SNP prediction (SIFT, PolyPhen-2) are widely accepted but prone to false positives/negatives. Including a discussion of their limitations and the validation steps taken (if any) would enhance the scientific rigor of this section.

Due to the limitations of some bioinformatics programs in assigning the effect of SNP-type mutations on the structure and function of mutated proteins and recognizing the limitations of these softwares in giving false negative or positive results. The bioinformatics prediction of the effect of the mutations was performed using multiple softwares such as: PolyPhen, SIFT, PhD SNP and SNP&GO, to increase the number of analyses and reduce the risk of reporting false positive or negative results. The result of these bioinformatics programs can be seen in Table 1. Additionally, the softwares I-Mutant 3.0, MUpro, mutation3D, Hope Project, MutPred2, TM-Aling and Ramachandran Plots were also used. To verify the effect of the mutations on the protein and to have a more complete set of bioinformatics programs.

 Page 7, Line 109-110: The use of I-Mutant 3.0 and MUpro for stability predictions is helpful, but the manuscript would benefit from discussing any experimental validation of these computational findings, as in silico results can differ from in vitro data.

This research project aims to analyze patients with high-grade gliomas by sequencing a panel of 324 genes and a bioinformatic analysis of the effect of mutations on the P53 and IDH1/2 proteins. Unfortunately, the project does not have the financial resources to carry out experimental validations on the effect of the mutations. However, in the future we will obtain resources to carry out the second experimental phase.

  • Page 8, Line 140-145: In the structural comparison between normal and mutated residues, the manuscript should provide more detailed data on the magnitude of changes in protein function (e.g., quantitative metrics for TM-score/RMSD) to give clearer insights into the biological impact of these mutations.

The data is provided in the following sections:

Data on TM-score/RMSD are provided in detail in Title 3.8 and Table 5.

3.8. Structural Differences between Normal and Mutated Residues Using TM-Aling and Ramachandran Plots

We ran the TM-aling and SWISS-MODEL programs to further analyze the changes between the structures of the wild-type proteins and the homology-modeled mutated proteins. The lower the TM score and the higher the RMSD value, the greater the difference between the structures. On the contrary, it may also be the case that the TM scores may be high as well as the RMSD value. Furthermore, the higher the score of the Ramachandran graph, the more the structure will be favored (Table 5), it was only shown that the C238W polymorphism for TP53 affects the structure of the protein, while for the IDH1 protein it was found that R132H was the most deleterious SNP, having the low TM score and higher RMSD, with a significant score in favor of the Ramachandran plot. This scenario is different with the non-synonymous SNP of IDH2 as it has a high TM score and a very low RMSD score.

Table 5. TM-matching score, RMSD value and Ramachandran plot, a) a score of 1 indicates a perfect match between the mutated and wild-type protein structures b) a higher RMSD score meant more structural differences between the two proteins compared. c) the Ramachandran plot score is calculated in percentage and provides an idea of ​​how well two structures seem to compare to each other.

Gene

nsSNP

TM-aling

SWISS-MODEL

TM-score a

Value RMSD b

Ramachandran graphic score c

TP53

Y236C

1.00

0.003

95.40%

TP53

V274G

1.00

0.004

95.40%

TP53

V157F

1.00

0.041

95.40%

TP53

C238W

0.999

 0.053

93.86%

TP53

A161S

1.00

0.001

95.40%

TP53

T125M

1.00

0.002

95.40%

TP53

R248Q

1.00

0.002

95.40%

TP53

R248W

1.00

0.001

95.40%

TP53

Y220H

1.00

0.002

95.40%

TP53

R175H

1.00

0.008

95.40%

TP53

C124T

1.00

0.006

95.40%

TP53

H179Q

1.00

 0.002

95.40%

TP53

G279E

1.00

0.004

95.40%

IDH1

R132H

0.8382

3.730

96.33%

IDH2

R172K

1.00

0.004

96.44%

  • Page 10, Line 198-203: The manuscript concludes that certain SNPs in TP53 and IDH1/2 are "deleterious," but the threshold for determining this isn’t explained. Specify how cutoff values for PolyPhen-2, SIFT, and other tools were chosen for defining deleteriousness.

The evaluator is right, the MSC SCORE values ​​for the TP53, IDH1 and IDH2 genes are attached according to the program used (PolyPhen2 or SIFT).

Additionally, these values ​​appear in Table 1, columns 7 and 9. Where the MSC_SCORE is compared vs the value of the mutation predicted by PolyPhen2 or SIFT.

Additionally, we include this paragraph in the results:

The PolyPhen-2 and SIFT software were runner and the thresholds were determinate using the MSC (mutation significance cutoff) program, available at (https://www.hgid.org/computational-tools/).The mutation significance cutoff (MSC) was to TP53 (PolyPhen2=0,033 and SIFT=0,133), IDH1 (PolyPhen2=0,823 and SIFT=0,995), and IDH2 (PolyPhen2=1,000 and SIFT=0,995), this score was use like thresholds to analyses later. we determined that 13 non-synonymous SNPs for the TP53 gene were damaging with the PolyPhen-2 tool, while for the SFIT software, only 11 were classified as harmful, the non-synonymous SNPs (A161S, C124T) showed scores greater than 0.00 and are reported as tolerable mutations (Table 1). The same happened with the IDH1 SNP, which is tolerated for SIFT and damaging for PolyPhen-2. For the IDH2 polymorphism, this was harmful for both predictors.

  • Page 12, Line 285-287: The manuscript mentions alterations in metal-binding properties due to SNPs but lacks an explanation of how these alterations mechanistically affect protein function. Provide more insight into how these changes contribute to gliomagenesis or tumor progression.

The reviewer is right, so we add the following paragraph to the discussion:

The P53 protein depends on certain endogenous metals for its function, especially on Zinc (Zn) ions [x1]. Structural studies have demonstrated the presence of Zn ions in its active site [x2] and the importance of these metal ions in the following functions has been verified: the binding of the DBD domain (DNA Binding Domain) to DNA occurs only in the presence of Zn, the absence of Zn decreases the activity of the protein, the DNA binding domain is stabilized by Zn, the interaction between P53-DNA is altered by mutations in the DNA binding domain, mutations in the DBD domain lead to misfolding of the protein and poor binding to Zn ions, mutations in the DBD domain lead to poor transcriptional activity of the P53 protein, in the absence of Zn the binding to DNA is considered less stable. Additionally, it has been discovered that Zn can help reactivate mutated proteins and that Zn sequestration leads to an inactivation of the P53 protein [x3-x15]. This highlights the importance of the P53 mutations (Y220H=27%, R175H=67%, and H179Q=72%), which alter to a certain extent the binding of metals to the P53 protein and could lead to all these alterations in the function of the protein.

On the other hand, it has been observed that divalent metal ions such as magnesium and manganese (Mg2 and Mn2) when binding to the substrate (isocitrate) of the IDH1 protein enhance the catalytic activity of the protein [x16]. The R132H mutation represents about 80% of the mutations found in this gene and reduces the activity of the wild-type IDH1. But confers a gain of function in the mutated proteins to produce oncometabolites such as (2-hydroxyglutarate) [x16]. The IDH1 mutation (R132H = 55%) by reducing the metal binding of the protein could affect the catalytic activity of the wild-type IDH1 by converting isocitrate to α-ketoglutarate in the cytoplasm.

The association between IDH2 gene mutations and metal ions has been little explored. However, due to the similarity in amino acid sequence, shared domains and three-dimensional structure with respect to IDH1, it could be suggested that their metabolic activity would be affected in a similar way. The IDH2 mutation (R172K=40%) by reducing the metal binding of the protein could affect the catalytic activity of the wild-type IDH2 by converting isocitrate to α-ketoglutarate in the mitochondria.

References

  1. Alfadul SM, Matnurov EM, Varakutin AE, Babak MV. Metal-Based Anticancer Complexes and p53: How Much Do We Know? Cancers. 19 de mayo de 2023;15(10):2834.
  2. Cho Y, Gorina S, Jeffrey PD, Pavletich NP. Crystal Structure of a p53 Tumor Suppressor-DNA Complex: Understanding Tumorigenic Mutations. Science. 15 de julio de 1994;265(5170):346-55.
  3. Structural Biology of the Tumor Suppressor p53 and Cancer‐Associated Mutants. En: Advances in Cancer Research [Internet]. Elsevier; 2007 [citado 17 de septiembre de 2024]. p. 1-23. Disponible en: https://linkinghub.elsevier.com/retrieve/pii/S0065230X06970018
  4. Butler JS, Loh SN. Structure, Function, and Aggregation of the Zinc-Free Form of the p53 DNA Binding Domain. Biochemistry. 1 de marzo de 2003;42(8):2396-403.
  5. Bullock AN, Henckel J, Fersht AR. Quantitative analysis of residual folding and DNA binding in mutant p53 core domain: definition of mutant states for rescue in cancer therapy. Oncogene. 2 de marzo de 2000;19(10):1245-56.
  6. Loh SN. The missing Zinc: p53 misfolding and cancer. Metallomics. 2010;2(7):442.
  7. Pavletich NP, Chambers KA, Pabo CO. The DNA-binding domain of p53 contains the four conserved regions and the major mutation hot spots. Genes Dev. 1 de diciembre de 1993;7(12b):2556-64.
  8. Joerger AC, Fersht AR. Structure–function–rescue: the diverse nature of common p53 cancer mutants. Oncogene. 2 de abril de 2007;26(15):2226-42.
  9. Freed-Pastor WA, Prives C. Mutant p53: one name, many proteins. Genes Dev. 15 de junio de 2012;26(12):1268-86.
  10. Kogan S, Carpizo DR. Zinc Metallochaperones as Mutant p53 Reactivators: A New Paradigm in Cancer Therapeutics. Cancers. 29 de mayo de 2018;10(6):166.
  11. Blanden AR, Yu X, Wolfe AJ, Gilleran JA, Augeri DJ, O’Dell RS, et al. Synthetic Metallochaperone ZMC1 Rescues Mutant p53 Conformation by Transporting Zinc into Cells as an Ionophore. Mol Pharmacol. mayo de 2015;87(5):825-31.
  12. Chen Y, Gao T, Wang Y, Yang G. Investigating the Influence of Magnesium Ions on p53–DNA Binding Using Atomic Force Microscopy. Int J Mol Sci. 21 de julio de 2017;18(7):1585.
  13. Méplan C, Richard MJ, Hainaut P. Metalloregulation of the tumor suppressor protein p53: zinc mediates the renaturation of p53 after exposure to metal chelators in vitro and in intact cells. Oncogene. 2 de noviembre de 2000;19(46):5227-36.
  14. Butler JS, Loh SN. Zn2+-Dependent Misfolding of the p53 DNA Binding Domain. Biochemistry. 1 de marzo de 2007;46(10):2630-9.
  15. Hainaut, P, Milner, J. A structural role for metal ions in the “wild-type” conformation of the tumor suppressor protein p53. 15 de abril de 1993;(53):1739-42.
  16. Liu S, Abboud MI, John T, Mikhailov V, Hvinden I, Walsby-Tickle J, et al. Roles of metal ions in the selective inhibition of oncogenic variants of isocitrate dehydrogenase 1. Commun Biol [Internet]. 1 de noviembre de 2021 [citado 17 de septiembre de 2024];4(1).

  • Page 14, Line 355-357: The mutation A161S is described as novel, but its pathogenic potential is conflicting. It would be beneficial to suggest potential experimental validation steps, such as in vitro functional assays, to determine whether this SNP indeed impacts protein function. 

The following paragraph is attached to the discussion:

In our study, we report on a polymorphism (A161S) not yet classified in the databases; this represents a SNP with conflicting interpretation, since there are predictors, such as SIFT that classify it as tolerable. Therefore, experimental validation is required, e.g., an in-vitro functional assay, to determine whether this SNP impacts protein function.

We thank the reviewer for his valuable contributions that helped to greatly improve the manuscript.

Reviewer 2 Report

Comments and Suggestions for Authors

This study reported new single nucleotide polymorphisms (SNPs) in glioma patients, of which their pathogenicity, and how they affect the function and stability of the protein. However, the study has some issues that need further confirmation.

I have only a few minor suggestions for the authors to consider:

1. In Table 2, the polymorphism of G279E is in conflict between the programs I-Mutant 3.0 and MUpro, which program should prevail?

2. In Figure 1, can the mutation site of the protein be enlarged?

3. In figure 1, please indicate in the notes to the figure what software was used to produce it.

4. In table 4, please indicate the headings in a,b,c.

5. What are the reasons for the difference between IDH1 and IDH2 in Table 5?

6. Please verify the changes in TP53 and IDH1/2 in these glioma patients with relevant experiments.

7. Please verify what role TP53 and IDH1/2 play in the relevant pathway.

8. Are TP53 and IDH1/2 also variable in prognostic glioma patients?

9. Are there drugs that target TP53 and IDH1/2 to treat glioma patients?

Author Response

This study reported new single nucleotide polymorphisms (SNPs) in glioma patients, of which their pathogenicity, and how they affect the function and stability of the protein. However, the study has some issues that need further confirmation.

I have only a few minor suggestions for the authors to consider:

  1. In Table 2, the polymorphism of G279E is in conflict between the programs I-Mutant 3.0 and MUpro, which program should prevail?

The results of the I-Mutant V3.0 bioinformatics program should prevail, since the other specific amino acid changes tend to decrease the stability of the protein. Therefore, a similar effect would be expected in the G279E mutation.

  1. In Figure 1, can the mutation site of the protein be enlarged?

Yes, it is possible, but the intention of the figure is to show the total structure of the protein and the location of the mutation. The enlarged location of the sample is shown in Table 3, where an enlargement of all the samples is performed and the structural effect of the amino acid change is shown.

  1. In figure 1, please indicate in the notes to the figure what software was used to produce it.

The software used to make the figure was included in the description of the figure.

  1. In table 4, please indicate the headings in a,b,c.

The headings were indicated as 4a, 4b and 4c. In this way, the legend and the table correspond.

  1. What are the reasons for the difference between IDH1 and IDH2 in Table 5?

Because IDH1 and IDH2 are different proteins. IDH1 acts in the cytosol of the cell and IDH2 in the mitochondria. For this reason, they are treated as different proteins in the table.

  1. Please verify the changes in TP53 and IDH1/2 in these glioma patients with relevant experiments.

This research work is an approximation using next-generation sequencing, computational biology and bioinformatics. In the future, an experimental project will be submitted for funding to verify the effect of these mutations at an experimental level in the laboratory. However, for now, the experiments are not part of this computational research.

  1. Please verify what role TP53 and IDH1/2 play in the relevant pathway.

The interactions and metabolic pathways in which the mutated proteins participate are shown in Table 6. In addition, their importance in the discussion is discussed.

  1. Are TP53 and IDH1/2 also variable in prognostic glioma patients?

According to the literature, the TP53 protein is strongly related to a poor prognosis in patients with gliomas and a decrease in sensitivity to chemotherapy. In contrast, mutations in the IDH1 and IDH2 proteins are related to a good prognosis in patients with gliomas.

  1. Are there drugs that target TP53 and IDH1/2 to treat glioma patients?

The drug used to treat these patients is Temozolomide and radiotherapy. Final del formularioThese two treatments create bonds in the DNA that generate breaks in the DNA and lead to cell death. On the other hand, the enzyme MGMT seems to reverse the effect of temozolomide by correcting these bonds. Currently, drug VORASIDENIB and IVOSIDENIB, which has IDH1/2 as its molecular target, have been described for the treatment of gliomas.

Note: Changes in Yellow are highlighted in the document

Round 2

Reviewer 1 Report

Comments and Suggestions for Authors

Revised manuscript has improved significantly